# Experimental Investigations into the Pyrolysis Mechanism and Composition of Ceramic Precursors Containing Boron and Nitrides with Different Boron Contents

**DOI:** 10.3390/ma15238390

**Published:** 2022-11-25

**Authors:** Yiqiang Hong, Guoxin Qu, Youpei Du, Tingting Yuan, Shuangshuang Hao, Wei Yang, Zhen Dai, Qingsong Ma

**Affiliations:** 1Science and Technology on Advanced Ceramic Fibers and Composites Laboratory, College of Aerospace Science, National University of Defense Technology, Changsha 410073, China; 2Beijing System Design Institute of Mechanical-Electrical Engineering, Beijing 100871, China; 3The Fourth Academy of CASIC, Beijing 100028, China

**Keywords:** ceramic containing boron and nitrides, liquid ceramic precursor, moisture stability, thermo-oxidative stability

## Abstract

In this work, a novel ceramic precursor containing boron, silicon, and nitrides (named SiBCN) was synthesized from liquid ceramic precursors. Additionally, its pyrolysis, microstructure, and chemical composition were studied at 1600 °C. The results showed that the samples with different boron contents had similar structural composition, and both of the two precursors had stable amorphous SiBN structures at 1400 °C, which were mainly composed of B-N and Si-N and endowed them with excellent thermo-oxidative stability. With the progress of the heating process, the boron contents increased and the structures became more amorphous, significantly improving the thermal stability of the samples in high-temperature environments. However, during the moisture treatment, the introduction of more boron led to worse moisture stability.

## 1. Introduction

The development of a new generation of supersonic aircraft has stimulated the demand for a new generation of high-temperature thermal protection materials [1,2,3,4], which need to have excellent oxidation resistance and ablation resistance under ultrahigh temperatures, high stresses, and oxidizing–corrosive atmospheres [5].

Ceramics are defined as inorganic, nonmetallic materials made up of ionic, covalent, or mixed bonds [6], and nitride ceramics are kinds of ceramic compounds connected by an inorganic hybrid element-N chemical bond. Nitride ceramics have attracted much attention because of their unique properties [7,8]. The introduction of a nitrogen atom into ceramic structures has significant effects in enhancing their wear, toughness, mechanical properties, and thermo-oxidative stability [9]. In general, nitride ceramics have been synthesized as powders and as thin films [10,11]. Wen et al. reported high-entropy nitride ceramics, and their results showed that the nitride ceramics had excellent thermal stability [12]. In addition, nitride ceramics are also widely used in the fields of energy storage, wear-resistant coatings, and diffusion barriers for microelectronics [13,14,15]. According to the extensive literature reports [6,16,17], the existence of boron atoms also endows the ceramic materials with excellent thermal conductivity (60–120 W/Mk), low electrical resistivity (9–33 μΩcm), and high elastic modulus (250–560 GPa). Xiang et al. [18] investigated the ablation behavior of ZrB_2_-SiC/BN ceramics, and the results showed that boride ceramics exhibited excellent ablation resistance and configuration stability. Zhang et al. [19] reported the ablation behavior of hot-pressed ZrB_2_–20 vol% SiC using arc-jet testing with heat fluxes of 1.7 mW/M^2^ and 5.4 mW/m^2^ under different heating temperatures. During the heat treatment, the temperature of the ceramic surface was between 1700 and 2300 °C, and the results showed that the ZrB_2_-based composite possessed better thermo-oxidative and configurational stability. Based on these studies, the incorporation of nitrogen and boride atoms into ceramic materials possesses great potential to achieve superb performance in the field of ablation materials. At present, ceramic materials containing elemental boron, silicon, and nitrogen (named SiBCN) are commonly prepared via magnetron sputtering deposition and mechanochemical synthesis [20,21]. However, these methods are not conducive to industrial processing, greatly limiting the application of ceramic materials containing boron and nitrogen atoms. The development of new synthesis methods of SiBCN is still a major challenge to researchers, not to mention the systematic study of the pyrolysis mechanisms of liquid precursors.

In this work, we successfully synthesized the ceramic precursors with different boron contents via the polymer precursor method, and then we systematically explored the differences in the intrinsic structure of the precursors and the pyrolysis evolution under different environmental conditions for the first time. This work not only provides a method for preparing SiBCN but also extends its application to ablation materials.

## 2. Experimental

### 2.1. Materials and Methods

Methylhydrodichlorosilane was purchased from Alpha Chemical Co., Ltd., Shijiazhuang, China. Boron trichloride and hexamethyldisilazane were purchased from Beijing Chemical Factory, Beijing, China. N-hexane was obtained from the Tianjin Chemical Reagent Co., Tianjin, China. All of the reagents were used without further purification. Boron trichloride was dissolved in N-hexane, and the three raw materials were mixed according to a certain proportion. The system was heated to a predetermined temperature, kept constant for a period of time, and then the temperature was reduced to obtain the SiBCN precursor (polyborosilazane (PBSZ), Mw = 7142, Mw/Mn = 3.66) [22]. When changing the feed ratio of the boron source, the sample with higher boron content was named 15 MB, while that with lower boron content was named as 15 M. The 15 M and 15 MB were placed in a tube furnace, which was then filled with ammonia. According to a certain procedure, the target ceramic products were obtained by heating up to 400, 600, 800, 1000, 1200, 1400, and 1600 °C and maintaining the temperature for 2 h.

### 2.2. Measurements

Fourier-transform infrared (FTIR) spectroscopy measurements were performed on a Tensor 27 spectrometer (Bruker Corp., Karlsruhe, Germany), by using KBr disks at the ambient temperature. ^1^H-NMR and ^29^Si-NMR spectra were recorded on a Bruker Advance 400 MHz NMR spectrophotometer (Bruker Corp., St. Gallen, Switzerland). Thermogravimetric analysis (TGA) was carried out from the ambient temperature to 900 °C on a Netzsch STA409PC (Netzsch, Selb, Germany) at a heating rate of 10 °C/min in nitrogen and air atmospheres. Oxidation resistance tests were conducted in a muffle furnace at high temperatures under an air atmosphere. The fracture surface of the hybrids was observed on a Hitachi S-4800 scanning electron microscope (SEM, Hitachi Ltd., Tokio, Japan) at an accelerating voltage of 10 kV and on a transmission electron microscope (TEM, Talos F200X, FEI Company, Hillsboro, OR, USA). The crystal structure of the powder ceramics was characterized by X-ray diffraction (XRD, Bruker D8AA25, Karlsruhe, Germany) with Cu Kα radiation (=0.154 nm, 40 kV, and 40 mA). The diffraction patterns were scanned from 10° to 80° of 2θ in a step-scan mode at a step of 0.026° and a scanning speed of 2°/min. The elemental contents of the final ceramic products were determined through inductively coupled plasma optical emission spectrometry (ICPOES) (Thermo IRIS intrepid II).

## 3. Results and Discussion

### 3.1. Characterization of Nitride Ceramic Precursors

Two nitride ceramic precursors (15 M and 15 MB) with different boron contents were synthesized by similar chemical synthesis methods. Due to the higher feed ratio of boron-containing raw materials, the nitride ceramic precursor 15 MB had a higher boron content than 15 M. Nevertheless, both of them were uniformly dispersed in low-viscosity liquids at room temperature

As shown in Figure 1 (^1^H-NMR), the hydrogen chemical shift peaks belonging to Si-H (5.0~4.4 ppm) [23], Si(H)-CH_3_ (0.3–0.45 ppm) [24], and Si-(CH_3_) (0.1 ppm) appear on the two ^1^H-NMR spectra at the same time; the difference is that there is a more obvious Si-H chemical shift signal and a relatively weak signal peak of Si-(CH_3_) in the 15 M spectrum. In addition, a weak chemical shift signal of N-H (0.8 ppm) [25] can be observed in the 15 M spectrum.

The chemical structures of 15 M and 15 MB were examined by FTIR. Figure 2 shows the FTIR of 15 M and 15 MB; the characteristic absorption bands of N-H [26], C-H, Si-H [27], B-N [28], Si-CH_3_, and Si-N [29] can be observed at 3400 cm^−1^, 2900 cm^−1^, 2200 cm^−1^, 1400 cm^−1^, 1250 cm^−1^, and 1100~900 cm^−1^, respectively. Comparing the signal strength of the two precursors, it can be seen that the 15 MB precursor has weaker Si-H and Si-N signals, while B-N is significantly stronger than Si-N, indicating that it has a higher boron content. Combined with the characterization results of the ^1^H NMR and FTIR spectra, the two precursors have similar main structures and types of functional groups, and the broadening of the spectral peaks means that all kinds of functional groups have many different chemical environments, with a mixture of multiple local structures. A possible molecular structure of the ceramic precursors is shown in Figure 3.

### 3.2. Ceramic Process of Nitride Ceramic Precursors

The thermogravimetric curves and ammonia cracking yields of the 15 M and 15 MB precursors in nitrogen are shown in Figure 4. There are sharp declines in the weight curves of both of the precursors before 400 °C, and the curves reach a peak after 800 °C. When 400 °C, 600 °C, 800 °C, 900 °C, and 1000 °C are selected for ammonia cracking, the change trends of the ceramic yield are similar, but the ceramic yield of 15 M in NH_3_ is higher than that of 15 MB, indicating that the reaction between 15 M and ammonia is stronger than that of 15 MB. The results of elemental analysis show that the B and Si contents in the pyrolysis product of 15 M-900 °C are 2.4% and 44.0%, respectively, while, the B and Si contents in the pyrolysis product of 15 MB-900 °C are 12.7% and 34.7%, respectively, indicating the effectiveness of our synthesis strategy and that the different boron contents entail different pyrolysis mechanisms. It is interesting that the C contents of both of the samples are less than 0.6%, indicating that the pyrolysis process in NH_3_ can effectively remove the carbon components in the precursor. Next, the samples were prepared in different stages according to the ammonia cracking temperature, and the changes in composition and structure during the ceramic process were studied.

The structural evolutions of the pyrolysis products were characterized by FTIR, and the results are shown in Figure 5. There are a series of characteristic absorption peaks in the products of 15 M-400 °C, in which 3408 cm^−1^ belongs to the stretching vibration of N-H [30], 2960 cm^−1^ and 2900 cm^−1^ belong to the C-H bond of (Si-CH_3_) silyl methyl [31], 2135 cm^−1^ belongs to the Si-H [32] bond, the broad peak near 1400 cm^−1^ belongs to the B-N structure [33], 1261 cm^−1^ belongs to the characteristic bending vibration of silyl methyl, and the wide peak near 900 cm^−1^ belongs to the Si-N structure [34]. During the heating up to 600 °C, the N-H signal increased and broadened obviously, while the silyl-methyl-related signal nearly disappeared, indicating that the ammonolysis process and decarbonization reaction occurred during heat treatment. The Si-H bond disappeared, which may have been due to the coupling reaction between Si-H and Si-H or N-H, while the structure of B-N and Si-N still existed. When the temperature rose to 800 °C, the silyl methyl (Si-CH_3_) signal disappeared completely and the weak Si-H bond reappeared, which may have been due to the reduction of hydrogen caused by the partial decomposition of ammonia. There was no obvious difference in the pyrolysis products when the heat treatment reached 1000 ℃, indicating that the ceramic transformation of the precursor was completed at 800 °C.

Furthermore, solid ^29^Si-NMR was used to characterize the chemical environment evolution of Si units during pyrolysis, and the results are shown in Figure 6. The main signal (−20 ppm) [35] of the 15 M-400 pyrolysis product belongs to the silicon structural unit with methyl and hydrogen (SiCN_3_), and the weak signal at the high field (−35 ppm) [36] can be classified as the structure in which the silyl group is replaced by the amino group (SiHN_3_), which corresponds to the silyl methyl and silicon hydrogen signals in the FTIR. There is no Si-H structure in the FTIR of the 15 M-600 product. The signals at −21.22 ppm and −41.75 ppm in the silicon spectrum belong to SiCN_3_ and SiN_4_ structures, respectively [36]; the latter mainly indicates that an obvious ceramic transition has taken place, and the broadened weak signal near −90 ppm may come from the product of silicon coupling. There are only SiN_4_ structures in the pyrolysis products of 800 °C and 1000 °C, indicating that the ceramic transformation has been completed, which is consistent with the results of FTIR, and the main peak has a tendency to shift to the high field, which is due to the further removal of a small amount of carbon, accompanied by slight changes in the chemical environment of SiBCN.

The products of 15 MB-400 not only contain the structure of Si-H, but also have the signal of a SiC_3_N structure near 0 ppm, which is derived from the trimethylsilicon structure in 15 MB. The products at 600 °C are mainly composed of SiCN_3_ and SiN_4_. Compared with the 15 M pyrolysis products, the proportion of SiCN_3_ to SiN_4_ is higher, which is consistent with the residual silyl methyl signal in the infrared spectrum. The products of ammonia pyrolysis under heat treatment at 800 °C and 1000 °C are mainly SiN_4_ structures, and the signal has the trend of shifting to the high field. Unlike the 15 M products, there is a wide peak near −90 ppm in the 15 MB products, and the signal with the increase in temperature is particularly different from the signal (SiSiCN) [36] in the pyrolysis products of 600 °C, although they have the same peak position. This may be due to the existence of the Si-O structure introduced by the higher oxygen content of the 15 MB precursors.

The interesting thing about the above results is that the structural evolution trend during the process of ammonia cracking is similar. The main ceramic product is the SiBN amorphous network structure, which is mainly linked by B-N and Si-N. The introduction of more trimethylsilicon is used to increase the content of the boron source in 15 MB, which means more breakage of Si-C and the release of small carbon-containing gas molecules during the elevated heat treatment, resulting in lower ceramic yield, higher boron content, and B-N bonds. FTIR and solid ^29^Si-NMR spectra were used to characterize the structural evolution of the precursors during the ceramic process under different heat treatment temperatures.

### 3.3. Structure and Properties of Nitride Ceramics

#### 3.3.1. Composition and Thermal Stability of Ceramic Products

The initial samples derived from the two kinds of precursor ceramic product after heat treatment at 900 °C in NH_3_ were denoted as samples-900N, and they were used to study the composition evolution and thermal stability under different heat treatment temperatures in N_2_. The mass retention and element contents are shown in Table 1. During the process of high-temperature treatment, the weight loss of the 15 M ceramic products was relatively less. However, what stands out in Table 1 is the dramatic decline in the weight loss of both kinds of ceramic products when the heat treatment was above 1400 °C. With the increase in temperature, the table shows that there was a steady increase in the contents of elemental boron in both of the pyrolysis products. Considering the steady change in the quality retention rate of the whole sample, the loss of elemental boron during the high-temperature treatment was very small, indicating that the residual boron content in the amorphous structure has excellent thermal stability. The corresponding silicon content first increased and then decreased, indicating that the loss of silicon was very little when the heat treatment was lower than 1200 °C, and there was a sharp decline when the heat treatment reached to 1400 °C, which also led to the rapid decline in the corresponding quality retention rate of the whole sample.

XRD and FTIR spectra were used to explore the microstructural evolution of the pyrolysis products treated at different temperatures. As shown in Figure 7, the ceramic products of the two precursors remained amorphous when the heat treatment was below 1400 °C. However, the ceramic products of 15 M show a weak diffraction peak at 1500 °C; after heating up to 1600 °C, the diffraction peaks gradually become obvious. These peaks are assigned to α-Si_3_N_4_ and β-Si_3_N_4_, respectively. The unit cell [37] of α-Si_3_N_4_ is about twice that of β-Si_3_N_4_.

Compared with 15 M, the ceramic products of 15 MB have no obvious diffraction peak after 1500 °C heat treatment, but the diffraction peak becomes obvious when the heat treatment reaches 1600 °C. According the analysis of the signal of the diffraction peak, the ceramic products mainly contain more β-Si_3_N_4_ structure and less α-Si_3_N_4_ structure. Taking into account the differences in the crystallization temperature and the crystal size of the ceramic products, a greater content of the amorphous structure of 15 MB endows it with better thermal stability at high temperatures. In addition, due to more boron bringing more oxygen, a higher oxygen content was induced in the appearance of the Si_2_N_2_O crystal phase in the 15 MB ceramic products.

The FT-IR spectra of the high-temperature ceramic products are shown in Figure 8, which clearly shows the similar variation trends of the ceramic products with the increase in temperature. The N-H and Si-H absorption peaks of the samples-900N pyrolysis products effectively disappear after treatment at 1400 °C, and the residual signals are mainly B-N and Si-N absorption peaks. During the process of heating up to 1600 °C, the loss of Si units leads to the obvious weakening of the intensity of the Si-N signal. What stands out is the fact that the wide inclusion absorption peak becomes sharp, indicating that the structural regularity is improved, which is consistent with the crystallization phenomenon in the XRD spectrum. The difference between the two precursors is mainly reflected in the relative contents of the functional group types. The 15 MB contains more N-H bonds and B-N bonds, and the former disappear while the latter remain in the final product during high-temperature treatment.

The TEM diagrams of the ceramic products of the precursors under heat treatment at 1400 °C and 1600 °C are shown in Figure 9. The figure reveals that a similar amorphous structure is present in the ceramics product of the two precursors at 1400 °C, and the nanoscale crystallization regions (Region A and Region C) can only be seen locally in the high-resolution image, which is consistent with the XRD characterization results. After the treatment at 1600 °C, the ceramic products showed obvious granulation, and the high-resolution images showed lattice stripes in most areas, while the crystallization region reached dozens of nanometers, consistent with the strong diffraction peak in the XRD spectrum, which also indicated that the original amorphous structure of the ceramic products was destroyed and that crystallization occurred during the high-temperature treatment.

According to the above results, the ceramic products of the two precursors have good stability below 1400 °C and can maintain the SiBN amorphous network structure. When the temperature rises to 1600 °C, most of the boron atoms are retained while the silicon atoms are lost; there is granulation of the structure and morphology, and the high-temperature treatment promotes the formation of an ordered structure, ultimately resulting in the precipitation of SiN_4_ crystals. In particular, to compare the 15 M and 15 MB, the 15 MB ceramic products lose more weight but have more thermally stable amorphous structures.

#### 3.3.2. Moisture Stability of Ceramic Products

The main structures (Si-N and B-N) of the SiBCN ceramic products are moisture-sensitive. The pyrolysis products of the two precursors at different temperatures were placed in an airtight wet environment to study their moisture stability. The specific conditions were as follows: temperature 20 °C, humidity 90%, storage time 3 weeks. The physically adsorbed water was removed by low-temperature drying before detection. The relative mass and oxygen content after placement are shown in Table 2.

After heat treatment at different temperatures, we obtained the relevant samples, and we determined their moisture stability by assessing the quality of the samples before and after the moisture experiment. The table above illustrates the increase in their relative mass and oxygen content, indicating that the ceramic products react with environmental moisture and are oxidized. The relative change in the 15 M ceramic products was relatively small, and the moisture absorption of the pyrolysis products at 700 °C was the most obvious. However, high-temperature heat treatment (900~1200 °C) significantly reduced the variety and amounts of water-sensitive functional groups in the system, and the changes in relative mass and oxygen content were small. Compared with 15 M, the moisture stability of the 15 MB ceramic products was obviously poor; the relative mass increase was about 30%, and the oxygen content was also significantly increased, due to the higher boron content in 15 MB and its more easily hydrolyzed B-N structure.

The results of the FTIR spectra before and after the moisture treatment are shown in Figure 10. The main change lies in the appearance of Si-O bonds. Among the ceramic products of 15 M at the three temperatures depicted in the figure, there are obvious Si-O bonds in the pyrolysis products at 700 °C, and this trend decreases with the increase in temperature, indicating that the moisture stability is improved, which is consistent with the changes in the relative mass and oxygen content mentioned above. There are obvious Si-O bonds in the ceramic products of 15 MB after moisture treatment at all three temperatures, and the details of the spectra change significantly, indicating that an obvious reaction has taken place in the moisture environment. Through the above analysis, it can be seen that compared with the two precursors, the moisture stability of the 15 MB ceramic products is poor, and it is necessary to control the relative humidity in the air during their use.

### 3.4. High-Temperature Properties and Chemical Reaction Mechanism of the Nitride Matrix

#### 3.4.1. Air Oxidation of Ceramic Products

Samples-900N were used to explore the oxidation behavior under different heat treatment temperatures in air. The mass retention and the elemental contents are shown in Table 3. After the air oxidation treatment, the two kinds of ceramic products had high-quality retention. This table clearly shows the similar values of the boron and silicon contents during the heat treatments. The increase in oxygen content contributes to the stability of the overall quality. The high-quality retention rate shows that oxidation does occur and that more oxygen is introduced during the process of air heat treatment.

The XRD spectra of the ceramic products after oxidation are shown in Figure 11. An obvious α-Si_3_N_4_ [38] crystal phase appears after the 15 M pyrolysis products are oxidized by air at 1200 °C, and a SiO_2_ crystal phase appears after oxidation at 1400 °C. In comparison with the sample under heat treatment at 1500 °C in N_2_, the differences in the crystallization and oxidation behavior of the crystal phase present in the 15 M in N_2_ and air indicate that obvious changes have taken place in the ceramic structure during the air treatment. Unlike 15 M, the 15 MB pyrolysis products mainly remain amorphous even after air oxidation at 1400 °C, with only a weak SiO_2_ diffraction peak [39], which shows that the 15 MB ceramic products have better amorphous stability under the condition of air oxidation.

The ceramic products were analyzed by FTIR spectroscopy. As shown in Figure 12, the change trend of the ceramic products of the two precursors was similar. The N-H bond in the pyrolysis products of samples-900N gradually disappeared with the increase in temperature. The Si-H bond increased at first and then weakened, and the increase in its relative strength may have been due to its reaction with water and N-H bonds in the process of air oxidation at 1200 °C. In addition, the Si-O absorption peak appeared after oxidation. Compared with those of 15 M, the 15 MB pyrolysis products had more N-H bonds and fewer Si-H bonds, which disappeared after oxidation at 1400 °C. In addition, the 15 M ceramic products had an obvious Si-O absorption peak after oxidation at 1200 °C, but the 15 MB ceramic products only appeared after oxidation at 1400 °C, showing that the 15 MB ceramic products also have better oxidation resistance under high-temperature heat treatments.

The air oxidation products of 15 MB were detected by ^29^Si NMR, and the spectrum is shown in Figure 13. The main signal was an SiN_4_ structure [40], and the Si-O signal [39] in the pyrolysis product at 900 °C was very weak, while the oxidation reaction occurred after the high-temperature air treatment, so the relative signal intensity of Si-O increased.

SEM was used to detect the structure of the ceramic products before and after oxidation. As shown in Figure 14, the pyrolysis products of the two kinds of precursors have relatively dense surfaces. After air treatment at 1400 °C, the micromorphology of the 15 M ceramic products changes obviously, and more spheres appear in the surface. Compared with 15 M, the morphology of the 15 MB ceramic products changes little—only the edges of small particles become smooth. The results of SEM show that the 15 MB ceramic products essentially remain amorphous and have better oxidation resistance at high temperatures.

#### 3.4.2. Vacuum Reaction of Ceramic Products

The samples-900N treated at high temperatures in vacuum were used to investigate the evolution of the reaction type under similar actual conditions. The mass retention and elemental contents are shown in Table 4. After vacuum treatment, the two kinds of ceramic products showed weight loss, and the mass retention rate was similar to that in a nitrogen atmosphere. The boron and silicon contents changed little; however, the oxygen content decreased significantly. This shows that oxygen-containing species are mainly lost in vacuum treatment.

The XRD spectra of the ceramic products after vacuum treatment are shown in Figure 15; the changes in the two kinds of ceramic products are similar—there is no crystal phase at 1200 °C, and there are relative weak diffraction peaks in 15 M, indicating that 15 M-1400 tends to be amorphous. In comparison with 15 M, there are weak Si_3_N_4_ peaks and strong diffraction peaks of elemental Si and SiC [41] present in the 15 MB-1400, meaning that vacuum extraction of nitrogen can trigger the decomposition reaction of Si_3_N_4_.

The ceramic products were analyzed by FTIR spectroscopy, as shown in Figure 16. The N-H bonds and Si-H bonds in the pyrolysis products of the samples-900N gradually disappeared with the increase in temperature, and the wide package-like absorption peak tended to sharpen after treatment at 1400 °C. In addition, the main change lies in the gradual decrease in the abundance of Si-N bonds relative to B-N bonds, which was more obvious in the 15 MB ceramic products. This shows that the elemental silicon may be lost. The SEM diagram of the ceramic products after vacuum treatment is shown in Figure 17. There are some changes in the morphologies of the two kinds of ceramic products after the vacuum treatment. Compared with the 15 M ceramic products, the edges of the two ceramic products are smoother, while the granulation of the 15 MB ceramic products is obvious. Combined with the reduction in the oxygen contents in the ceramic products and the crystal phase analysis in XRD, it can be seen that elemental oxygen may combine with silicon to form gaseous products that depart during vacuum treatment. At the same time, the crystallization and decomposition of Si_3_N_4_ will occur, resulting in the formation of elemental Si and a small amount of SiC. The changes in the 15 MB ceramic products are more obvious than those of the two precursors.

## 4. Conclusions

In summary, novel ceramics containing boron and nitrides were successfully obtained via a liquid ceramic precursor and its pyrolysis mechanism, and their structural evolution under elevated heat treatment was first investigated. The results show that differences in boron content do not change the main structural composition of the precursor, and the thermal stability and pyrolysis evolution process show that the boron content does not affect the structural evolution of the precursor in the ceramic process. The introduction of trifunctional N, trifunctional B, and tetrafunctional Si promotes the formation of more amorphous SiBN structures, which endow the system with superior thermo-oxidative stability. Although the introduction of more boron sources can improve the formation of amorphous structures in the system, due to the affinity of boron for oxygen and water too much boron is not conducive to the moisture stability of the product.

## Figures and Tables

**Figure 1 materials-15-08390-f001:**
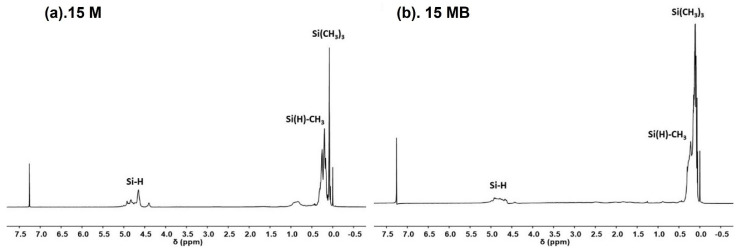
Structural characterization of 1H-NMR: (**a**) 15 M and (**b**) 15 MB.

**Figure 2 materials-15-08390-f002:**
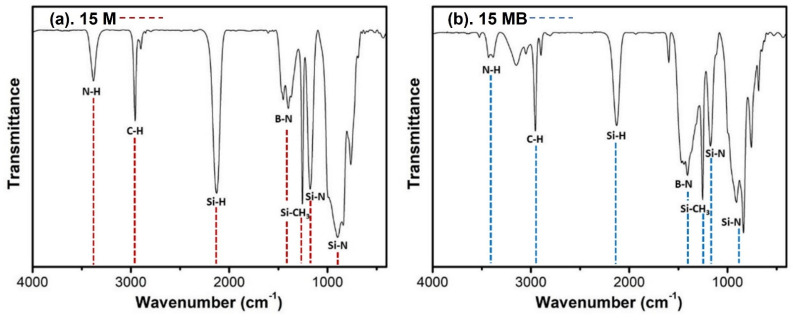
FTIR spectra of (**a**) 15 M and (**b**) 15 MB.

**Figure 3 materials-15-08390-f003:**
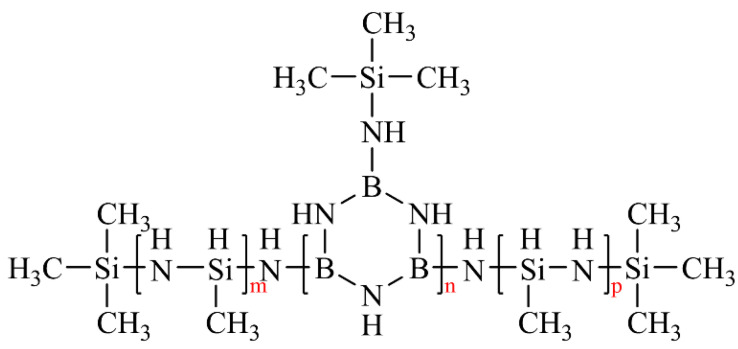
Possible molecular structure of ceramic precursors, and the m, n and p denote random degree of polymerization.

**Figure 4 materials-15-08390-f004:**
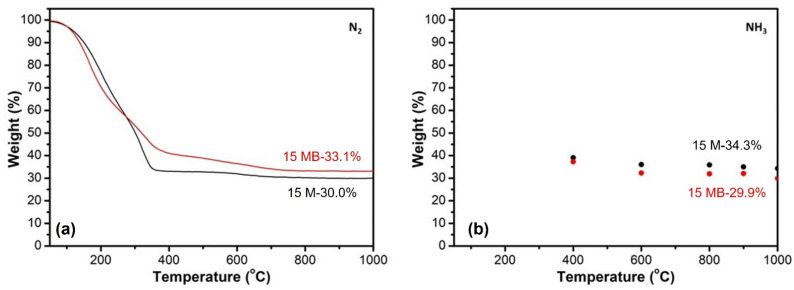
Thermal analyses and ammonia cracking yield of 15 M and 15 MB: (**a**) TGA in N_2_; (**b**) ammonia cracking yield.

**Figure 5 materials-15-08390-f005:**
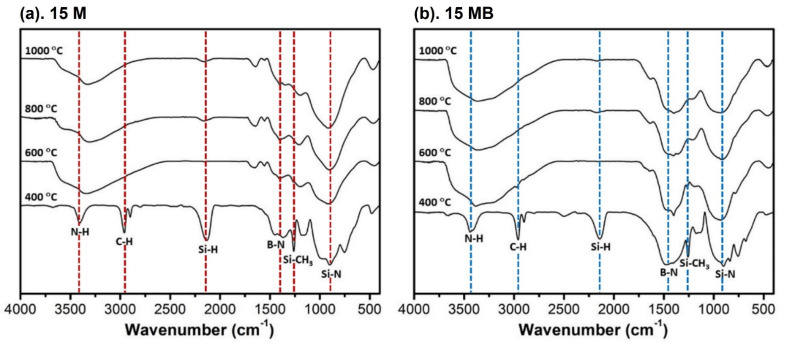
FTIR spectra of the pyrolysis products of (**a**)15 M and (**b**)15 MB at different temperatures.

**Figure 6 materials-15-08390-f006:**
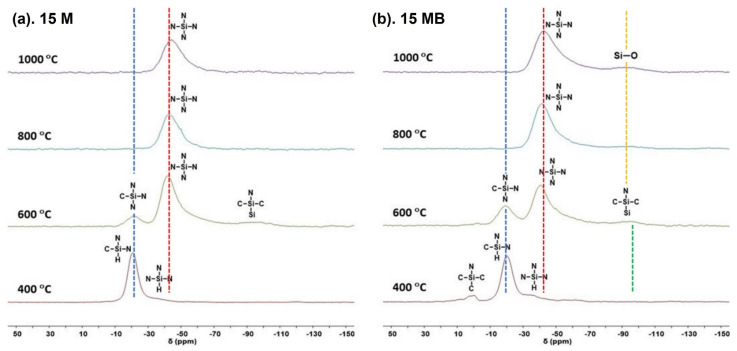
The ^29^Si-NMR spectra of the pyrolysis products of (**a**) 15 M and (**b**) 15 MB at different temperatures.

**Figure 7 materials-15-08390-f007:**
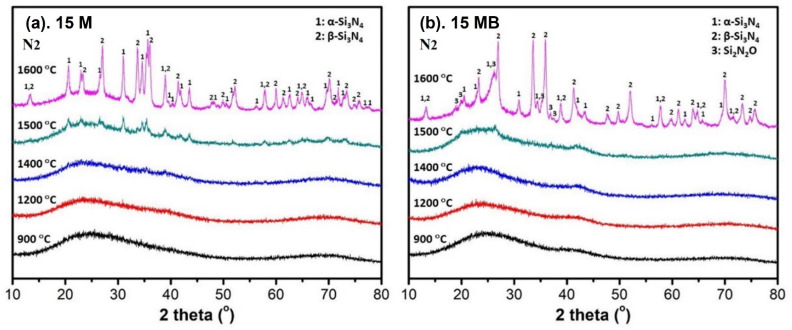
XRD spectra of the pyrolysis products of (**a**) 15 M and (**b**) 15 MB at different temperatures.

**Figure 8 materials-15-08390-f008:**
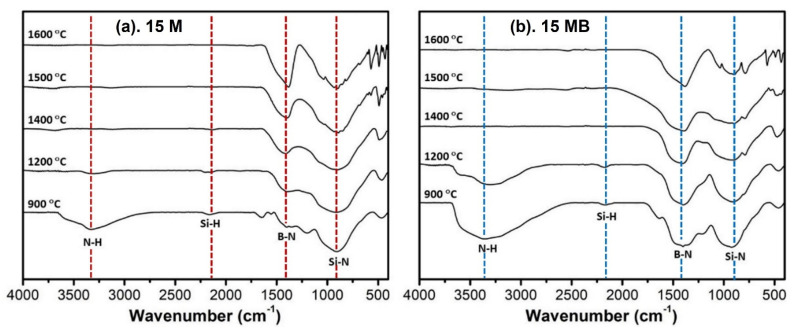
FT-IR spectra of the pyrolysis products of (**a**) 15 M and (**b**) 15 MB at 900~600 °C.

**Figure 9 materials-15-08390-f009:**
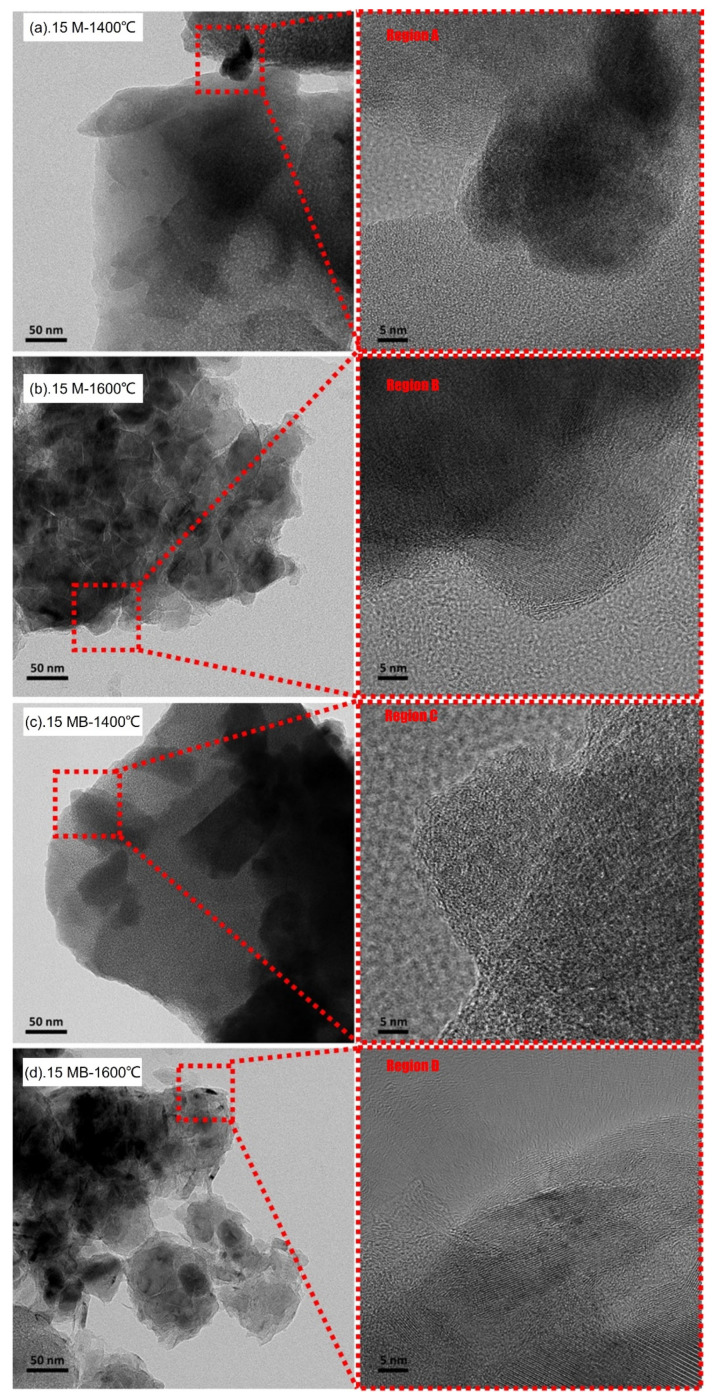
TEM images of the pyrolysis products of (**a**) 15 M, (**b**) 15 M, (**c**) 15 MB and (**d**) 15 MB at different temperatures.

**Figure 10 materials-15-08390-f010:**
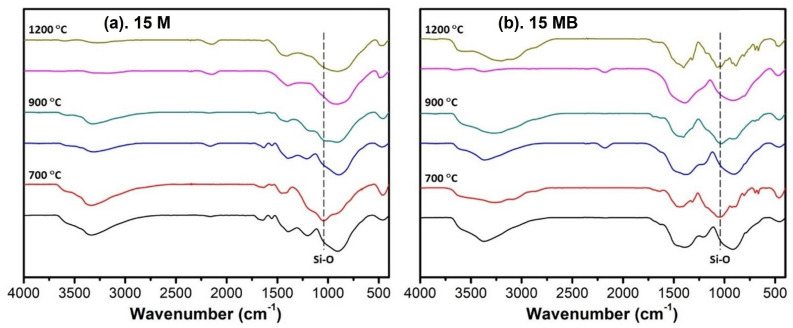
FTIR spectra of the moisture treatment products of the two kinds of (**a**) 15 M and (**b**) 15 MB precursors.

**Figure 11 materials-15-08390-f011:**
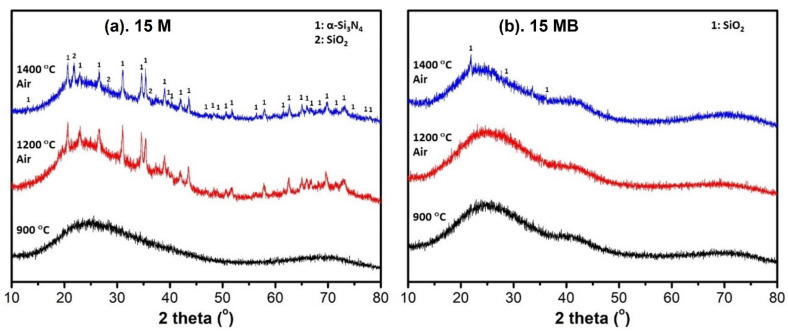
XRD spectra of the oxidation products of the two precursors (**a**) 15 M and (**b**) 15 MB at different temperatures.

**Figure 12 materials-15-08390-f012:**
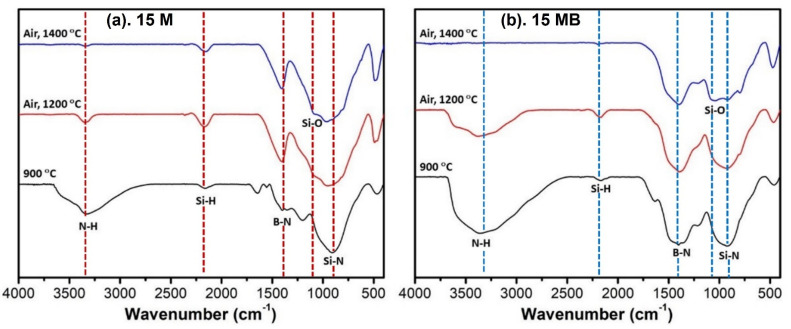
FTIR spectra of the oxidation products of the two precursors (**a**) 15 M and (**b**) 15 MB at different temperatures.

**Figure 13 materials-15-08390-f013:**
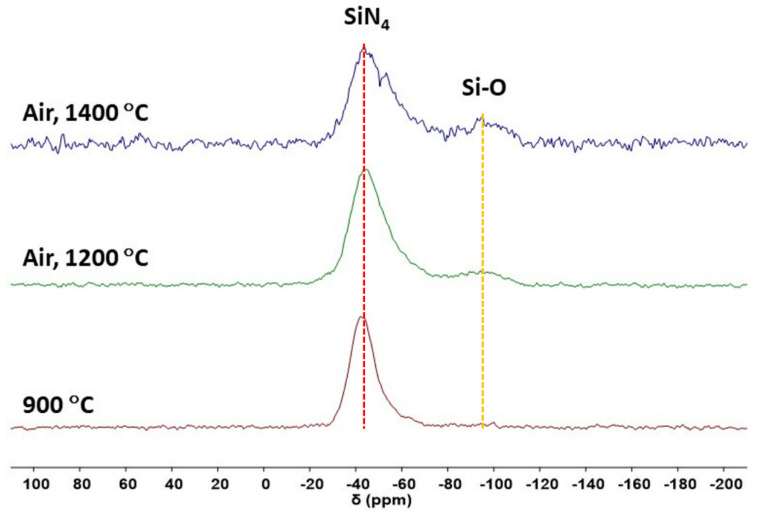
The ^29^Si-NMR spectra of 15 MB’s cracking and oxidation products.

**Figure 14 materials-15-08390-f014:**
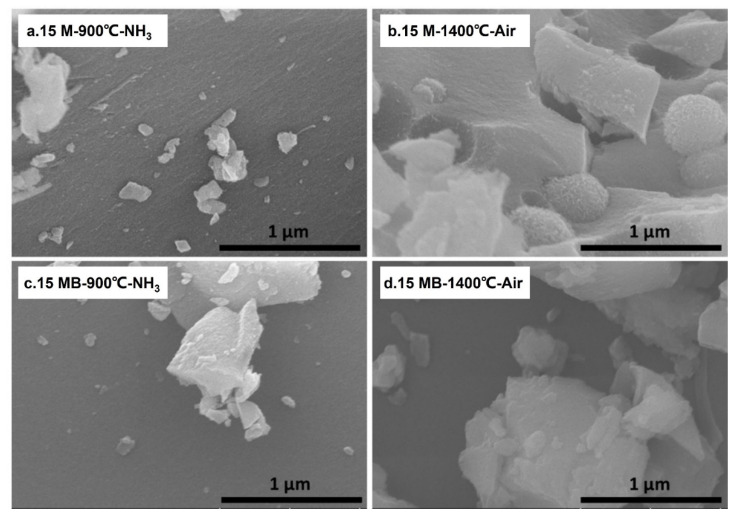
SEM diagrams of the air oxidation products of the two precursors (15 M (**a**,**b**) and 15 MB (**c**,**d**)).

**Figure 15 materials-15-08390-f015:**
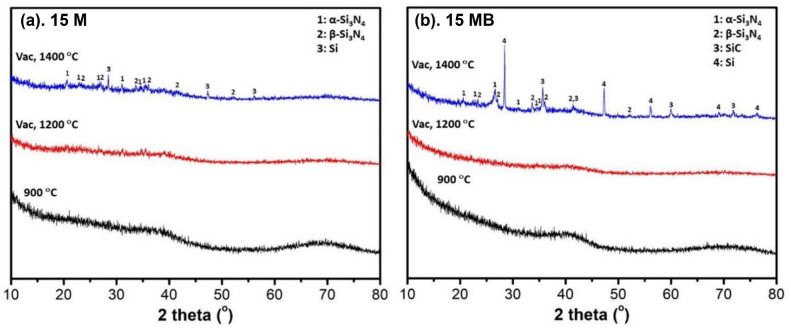
XRD spectra of vacuum-treated products of the two kinds of precursors at different temperatures (**a**) 15 M and (**b**) 15 MB.

**Figure 16 materials-15-08390-f016:**
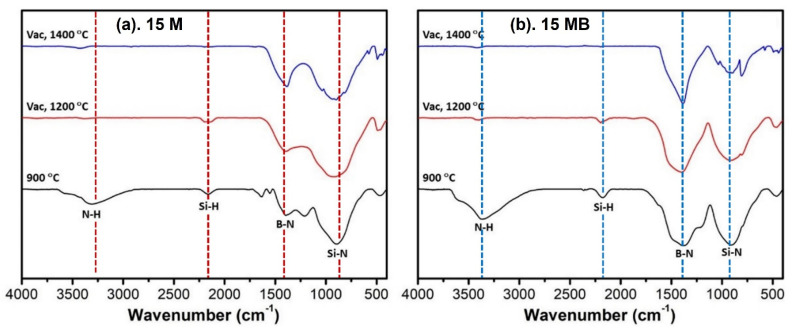
XRD spectra of the vacuum-treated products of the two kinds of precursors (**a**) 15 M and (**b**) 15 MB at different temperatures.

**Figure 17 materials-15-08390-f017:**
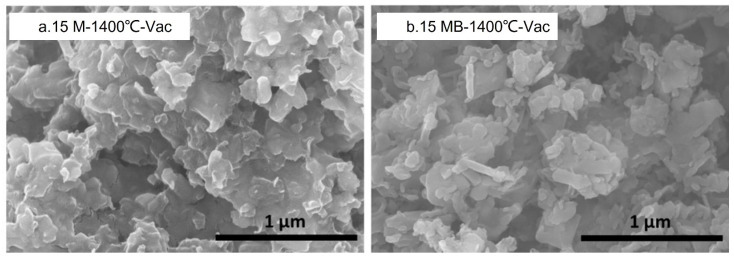
SEM diagrams of the vacuum treatment products of the two kinds of precursors (**a**) 15 M and (**b**) 15 MB.

**Table 1 materials-15-08390-t001:** High-temperature treatment of 15 M and 15 MB ceramic products.

Sample	T( °C)	Mass (wt%)	B (wt%)	Si (wt%)
15 M	900	100	2.4	44.0
	1200	90.94	3.3	54.9
	1400	89.08	3.6	53.8
	1600	76.93	4.1	49.3
15 MB	900	100	12.7	34.7
	1200	90.22	14.1	38.3
	1400	83.41	13.6	37.0
	1600	70.69	17.2	34.1

**Table 2 materials-15-08390-t002:** Moisture stability of 15 M and 15 MB ceramic products (“PrO” means “prior O” and “AfO” means “after O”).

Sample	T (°C)	Mass (wt%)	PrO (wt%)	AfO (wt%)
15 M	700	116.19	/	11.4
	900	115.93	5.0	12.7
	1200	106.50	3.7	7.3
15 MB	700	133.47	/	40.1
	900	128.92	5.9	22.7
	1200	137.61	3.6	34.6

**Table 3 materials-15-08390-t003:** Air oxidation of the 15 M and 15 MB ceramic products.

Sample	T (°C)	Mass (wt%)	B (wt%)	Si (wt%)	O (wt%)
15 M	900	100	3.2	49.2	7.1
	1200	95.79	3.2	50.6	9.4
	1400	94.96	3.1	50.7	12.0
15 MB	900	100	13.1	36.0	11.4
	1200	93.77	13.0	35.8	14.2
	1400	92.45	13.4	37.8	13.4

**Table 4 materials-15-08390-t004:** Vacuum treatment of 15 M and 15 MB ceramic products.

Sample	T (°C)	Mass (wt%)	B (wt%)	Si (wt%)	O (wt%)
15 M	900	100	3.2	49.2	5.03
	1200	91.41	3.4	54.9	1.67
	1400	87.64	3.6	54.6	1.86
15 MB	900	100	13.1	36.0	5.93
	1200	91.49	13.6	38.7	2.23
	1400	83.57	15.4	41.0	1.26

## Data Availability

Not applicable.

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
