# Peer review of "Experimental Investigations into the Pyrolysis Mechanism and Composition of Ceramic Precursors Containing Boron and Nitrides with Different Boron Contents"

_materials, 2022, doi:10.3390/ma15238390_

Round 1

Reviewer 1 Report

The study is very interesting from application point of view and deals with investigation of pyrolysis mechanism and composition of ceramic precursors. However, it is recommended to submit an article for review after a minimal check of content. I was surprised to see that in section 2. Experimental 2.1 Materials and preparation, the authors forgot to delete the information from instruction for authors template. The article can be published after implementing the following:

Line 38: please delete the dot from electrical resistivity: it should be μΩcm. I also prefer to use the international dimensions without submultiples (e.g.  Ωm).

Line 44: Please rephrase the sentence. I think there are multiple synthesis methods and the term major challenge is a little bit forced.

Line 58-72: These paragraphs should be deleted. You should rewrite the entire section 2.1

Line 76-84: please provide additional information related to the equipment used in measurements (Manufacturer, Country, Year, Version). Do this for all equipment and software used in your experiments.

Line 153: Please explain if the main signal acquired represents your results or is from current scientific literature

Author Response

Reviewer: 1

Q1: The paper refers to experimental investigations into the pyrolysis mechanism and composition of ceramics precursor containing boron and nitrides with different boron content. It presents interesting results, however the paper needs a careful re-reading. It looks like an internal draft than a submitted manuscript for review. 1.Parts from the “instructions to the authors” are left over in Materials & Preparation section (lines 58-72), 2.sentences are incomplete or difficult to understand (ex. “the nitride ceramic precursor has higher boron content than that”, in lines 89-90), and 3.instrumentation (TEM) is not included in the Measurements section, 4.Figure 7 is not mentioned in the text, 5.Table 1 in line 187 is mentioned as Table 2 and Table 3 is mentioned as Table 3.1 (line 268). 6.Furthermore, in all (except two) Figures the Y-axis has no title and the values quoted in all Tables are not explained in the text how they are obtained. Authors, should carefully re-read the manuscript and correct all the above-mentioned issues. 

 Responds:

Q1-1: We are very sorry, we have deleted the relevant description, revised the content of this part, and modified the title of this part according to your suggestion. Thank you again for your patient reminder.

Q1-2: Thank you for your reminder, we have modified the relevant description to make it easier for readers to understand.

        Due to the higher feed ratio of boron-containing raw materials, the nitride ceramic precursor (named as 15MB) has higher boron content than that (named as 15M).”

Q1-3: We have seriously added the description of the relevant instruments.

“Fracture surface of the hybrids was observed on a Hitachi S-4800 scanning electron microscope (SEM) at an accelerating voltage of 10 kV and transmission electron microscope (TEM, Talos F200X).”

Q1-4: We seriously added the description of the related graph.

“As show in the Figure 7, the ceramic products of the two precursors remain amorphous when the heat treatment is below 1400℃.”

Q1-5: Thank you for reminding us that we have modified the relevant description in the manuscript.

Q1-6: Thank you for your reminder, we have added the annotation and interpretation of spectrum and table data.

Q2: Furthermore, the section “Materials & Preparation” is limited into the discerption of the precursor preparation and does not include all the rest experimental details. Therefore, it is difficult for the reader to follow the experimental procedure. It may be better that this section is renamed into “Materials & Methods” and outline all the experimental procedures. Authors are also advised to clarify the “certain procedure” in line 56.

 Responds:

Thank you for your suggestion. According to your suggestion, we have revised the title and revised the relevant content in the manuscript as following:

“2.1 Materials & Methods

Methylhydrodichlorosilane was purchased from Alpha Chemical Co., Ltd, Shijiazhuang, China. Boron trichloride and hexamethyldisilazane were purchased from Beijing Chemical Factory, Beijing, China. N-hexane was obtained from the Tianjin Chemical Reagent Co., China. All the reagents were used without further purification. Boron trichloride is dissolved in N-hexane, and the three raw materials are mixed according to a certain proportion, the system is heated to a predetermined temperature, kept for a period of time, and then the temperature is reduced to get SiBCN precursor (polyborosilazane (PBSZ), Mw=7142, Mw/Mn=3.66) [21]. By changing the feed ratio of boron source, sample with higher boron content is named as 15MB, and that with low boron contents is named as 15M. The 15M and 15MB were placed in a tube furnace and then filled with ammonia. According to a certain procedure, the target ceramic products were obtained by heating up to 400, 600, 800, 1000, 1200, 1400, 1600 ℃ respectively and holding for 2 hours.”

Q2: The two graphs in Figure 1 should be better combined into one, for direct comparison. The same holds for figure 2 as well. In this case, the sentence “curves are shifted vertically for clarity” should be included in the figure captions. The same sentence should be included in all the rest figures, where multiple graphs are presented. When all graphs in a figure are not in the same scale, this should also be mentioned.

 Responds:

        Thank you for your suggestion. In order to facilitate the description of the article, we did not integrate the spectrum in the first place. Because the writing logic of the article has been determined, so in this revision, we use the format of data comparison to enhance the contrast of the data. In order to make it easier for readers to read, we have added additional notes to each comparison.

Q3:Authors in lines 216-219 state that “the nanoscale crystallization region can only be seen locally in the high-resolution image” and provide a TEM micrograph as evidence. I wonder why they have not included in Figure 9 Selected Area Electron Diffraction pictures, for a direct prof. SAE is a technique used to evaluate the sample's crystallinity, through diffused rings (for amorphous materials), rings composed by tinny discrete spots (poly/nano crystalline) or discrete spots in an ordered pattern (crystalline).

 Responds:

Thank you for your suggestion. In order to make it easier for readers to read, we mark the relevant region in the spectrum. In our labeling, we can clearly see that the crystal structure changes with the heating process. Thank you for your proposal. We will introduce your proposed SAE in the follow-up work to improve our work. Thank you again.

Q3:Authors in lines 237-238 state that “The relative mass and oxygen content after placement are shown in Table 2”. The last two columns in Table 2 are marked as “prO” and “afO”. Authors should also try to describe in the text the abbreviations used in Table 2. If “pr” and “af” mean “prior” and “after”, then the text in lines 237-238 should be rephrased.

 Responds:

Thank you for your suggestion. In order to make it easier for readers, we have marked the relevant tables.

“Table 2 Moisture Stability of 15M and 15MB Ceramic products. (“PrO” means “prior O”, and “AfO” means “after O”)”

Q4:Furthermore, 1.in line 246 authors state that “After 1200oC treatment, it is more stable to moisture”. On what evidence is this statement based, since Table 2 is limited up to 1200oC. 2.Similarly, in lines 119-120, authors state that “The main weight loss of the precursor is before 400°C”, without indicating where this weight loss is attributed.

 Responds:

Q4-1: Thank you for your suggestion. In order to make it easier for readers, we have modified the relevant description.

“The relative change of 15M ceramic products is relatively small, and the moisture absorption of pyrolysis products at 700°C is the most obvious. However, high temperature heat treatment (900~1200°C) significantly reduced the type and amount of water-sensitive functional groups in the system, the changes of relative mass and oxygen content are small. Compared with 15M, the moisture stability of 15MB ceramic products is obviously poor, the relative mass increase is about 30%, and the oxygen content is also significantly increased, which is due to the higher boron content in 15MB and more easily hydrolyzed B-N structure.”

Q4-2: Thank you for your suggestion. In order to make it easier for readers, we have modified the relevant description.

“There are sharp decline in the weight curves of both of the precursors before 400 ℃, and the curves reached a peak after 800 ℃.”

Q5: Comparing Figures 7, 11 & 15, it is observed a totally different shape for the 900oC curve. I assume that the 900oC curve is the reference in all cases, and the deviations are presented in each Figure compared to the 900oC curve. If this is the case, the reference curve should be the same for all the graphs. Authors should explain the different pattern.

 Responds:

The initial samples derived from the two kinds of precursor ceramic product after heat treatment at 900℃ in NH3.(named as sample-900N), sample-900N was sued to explore structure and properties of nitride ceramics. Thank you for your suggestion. In order to make it easier for readers, we have modified the relevant description. The reason for the difference pattern is that the sample is treated again in a different environment of 900℃.

Q6:  Comparing Tables 1, 2, 3 & 4, in some the 900oC is taken as the reference temperature (mass wt. 100%), in others (Table 2) there is no “reference” temperature, since at the lowest temperature (700oC) the mass wt. is 116.19%. This discrepancy confused the reader. A further confusion may arise combining the results presented in Tables with Figure 2. Clearly, the mass wt. (loss or uptake) is not 100% at the starting (reference) temperature (900oC). Authors should explain that in the text. 

  Responds:

After heat treatment at different temperatures, we get the relevant samples, and we explain the gas-moisture stability by weighing the quality of the samples before and after the moisture experiment.

Q7: Finally, the manuscript should be proof-read by a native English speaker, with materials science background. I am not a native English speaker myself, however I have spotted some obvious misuse of the English language. For example, in line 15 “were first studied at 1600oC”. Do authors mean that the study was carried “for the first time” and I believe that it is “up to” instead of “at”. I do not understand “These methods are not conductive” in line 43. These are just two examples – they are points in the manuscript that need re-phrasing.   

  Responds:

Thank you for your reminder. We have carefully revised the manuscript to make its language description meet the requirements of you and the journal.

Reviewer 2 Report

The paper refers to experimental investigations into the pyrolysis mechanism and composition of ceramics precursor containing boron and nitrides with different boron content. It presents interesting results, however the paper needs a careful re-reading. It looks like an internal draft than a submitted manuscript for review. Parts from the “instructions to the authors” are left over in Materials & Preparation section (lines 58-72), sentences are incomplete or difficult to understand (ex. “the nitride ceramic precursor has higher boron content than that”, in lines 89-90), and instrumentation (TEM) is not included in the Measurements section, Figure 7 is not mentioned in the text, Table 1 in line 187 is mentioned as Table 2 and Table 3 is mentioned as Table 3.1 (line 268). Furthermore, in all (except two)Figures the Y-axis has no title and the values quoted in all Tables are not explained in the text how they are obtained. Authors, should carefully re-read the manuscript and correct all the above-mentioned issues. 

Furthermore, the section “Materials & Preparation” is limited into the discerption of the precursor preparation and does not include all the rest experimental details. Therefore, it is difficult for the reader to follow the experimental procedure. It may be better that this section is renamed into “Materials & Methods” and outline all the experimental procedures. Authors are also advised to clarify the “certain procedure” in line 56.

The two graphs in Figure 1 should be better combined into one, for direct comparison. The same holds for figure 2 as well. In this case, the sentence “curves are shifted vertically for clarity” should be included in the figure captions. The same sentence should be included in all the rest figures, where multiple graphs are presented. When all graphs in a figure are not in the same scale, this should also be mentioned.

Authors in lines 216-219 state that “the nanoscale crystallization region can only be seen locally in the high-resolution image” and provide a TEM micrograph as evidence. I wonder why they have not included in Figure 9 Selected Area Electron Diffraction pictures, for a direct prof. SAE is a technique used to evaluate the sample's crystallinity, through diffused rings (for amorphous materials), rings composed by tinny discrete spots (poly/nano crystalline) or discrete spots in an ordered pattern (crystalline).

Authors in lines 237-238 state that “The relative mass and oxygen content after placement are shown in Table 2”. The last two columns in Table 2 are marked as “prO” and “afO”. Authors should also try to describe in the text the abbreviations used in Table 2. If “pr” and “af” mean “prior” and “after”, then the text in lines 237-238 should be rephrased.

Furthermore, in line 246 authors state that “After 1200oC treatment, it is more stable to moisture”. On what evidence is this statement based, since Table 2 is limited up to 1200oC. Similarly, in lines 119-120, authors state that “The main weight loss of the precursor is before 400°C”, without indicating where this weight loss is attributed.

Comparing Figures 7, 11 & 15, it is observed a totally different shape for the 900oC curve. I assume that the 900oC curve is the reference in all cases, and the deviations are presented in each Figure compared to the 900oC curve. If this is the case, the reference curve should be the same for all the graphs. Authors should explain the different pattern.

Comparing Tables 1, 2, 3 & 4, in some the 900oC is taken as the reference temperature (mass wt. 100%), in others (Table 2) there is no “reference” temperature, since at the lowest temperature (700oC) the mass wt. is 116.19%. This discrepancy confused the reader. A further confusion may arise combining the results presented in Tables with Figure 2. Clearly, the mass wt. (loss or uptake) is not 100% at the starting (reference) temperature (900oC). Authors should explain that in the text. 

Finally, the manuscript should be proof-read by a native English speaker, with materials science background. I am not a native English speaker myself, however I have spotted some obvious misuse of the English language. For example, in line 15 “were first studied at 1600oC”. Do authors mean that the study was carried “for the first time” and I believe that it is “up to” instead of “at”. I do not understand “These methods are not conductive” in line 43. These are just two examples – they are points in the manuscript that need re-phrasing.   

Author Response

Reviewer: 2

The study is very interesting from application point of view and deals with investigation of pyrolysis mechanism and composition of ceramic precursors. However, it is recommended to submit an article for review after a minimal check of content. I was surprised to see that in section 2. Experimental 2.1 Materials and preparation, the authors forgot to delete the information from instruction for authors template. The article can be published after implementing the following:

Q1: Line 38: please delete the dot from electrical resistivity: it should be μΩcm. I also prefer to use the international dimensions without submultiples (e.g.  Ωm).

  Responds:    

Thank you for your reminder, we have carefully revised the manuscript to make its description meet the requirements of you and the journal

Q2: Line 44: Please rephrase the sentence. I think there are multiple synthesis methods and the term major challenge is a little bit forced.

  Responds:

Thank you for your reminder, we have carefully revised the manuscript to make its description meet the requirements of you and the journal

Q3: Line 58-72: These paragraphs should be deleted. You should rewrite the entire section 2.1

 Responds:

Thank you for your suggestion. According to your suggestion, we have revised the title and revised the relevant content in the manuscript as following:

“2.1 Materials & Methods

Methylhydrodichlorosilane was purchased from Alpha Chemical Co., Ltd, Shijiazhuang, China. Boron trichloride and hexamethyldisilazane were purchased from Beijing Chemical Factory, Beijing, China. N-hexane was obtained from the Tianjin Chemical Reagent Co., China. All the reagents were used without further purification. Boron trichloride is dissolved in N-hexane, and the three raw materials are mixed according to a certain proportion, the system is heated to a predetermined temperature, kept for a period of time, and then the temperature is reduced to get SiBCN precursor (polyborosilazane (PBSZ), Mw=7142, Mw/Mn=3.66) [21]. By changing the feed ratio of boron source, sample with higher boron content is named as 15MB, and that with low boron contents is named as 15M. The 15M and 15MB were placed in a tube furnace and then filled with ammonia. According to a certain procedure, the target ceramic products were obtained by heating up to 400, 600, 800, 1000, 1200, 1400, 1600 ℃ respectively and holding for 2 hours.”

Q4: Line 76-84: please provide additional information related to the equipment used in measurements (Manufacturer, Country, Year, Version). Do this for all equipment and software used in your experiments.

  Responds:

Thank you for your reminder, we have carefully revised the manuscript to make its description meet the requirements of you and the journal

Q5: Line 153: Please explain if the main signal acquired represents your results or is from current scientific literature

  Responds:

Thank you for your reminder, we have carefully revised the manuscript and increased the literature citations of relevant signals.

Reviewer 3 Report

In their work "Experimental investigations into the pyrolysis mechanism and composition of ceramics precursor containing boron and nitrides with different boron content", the authors discuss composition of ceramics containing boron and nitrides.
This work fits with the journal scope. This work is short and has lots of flaws. However, there are major issues with this manuscript and must be addressed, and therefore, it cannot be accepted for publication in its current form:
• The entire manuscript needs to be re-written and authors should put much effort into meeting the journal format.
• The motivation for this work is not clear.
• The abstract is weak and requires more information and attention.
• The introduction is very broad and weak and requires additional information, in particular previous studies.

Author Response

Reviewer: 3

In their work "Experimental investigations into the pyrolysis mechanism and composition of ceramics precursor containing boron and nitrides with different boron content", the authors discuss composition of ceramics containing boron and nitrides.
This work fits with the journal scope. This work is short and has lots of flaws. However, there are major issues with this manuscript and must be addressed, and therefore, it cannot be accepted for publication in its current form:
Q1: The entire manuscript needs to be re-written and authors should put much effort into meeting the journal format.

  Responds:

Thank you for your reminder, we have carefully revised the manuscript to make its description meet the requirements of you and the journal

Q2: The motivation for this work is not clear.

  Responds:

Thank you for reminding us that we have carefully revised the manuscript and described the objectives of our work in the summary.

“The development of new synthesis methods of SiBCN is still a major challenge to researchers, not to mention the systematic study on the pyrolysis mechanism of liquid precursors.

In this work, we successfully synthesized the ceramics precursors with different boron content by the polymer precursor method, and systematically explored the differences of the intrinsic structure of the precursor and the pyrolysis evolution under different environmental conditions for the first time. This work not only provides a method for preparing SiBCN but also extends their application to alation material. “

Q3: The abstract is weak and requires more information and attention.

Responds:

Thank you for your reminder, we have carefully revised the abstract of the manuscript to meet the requirements of you and the journal.

“Abstract: In this work, a novel ceramic precursor containing boron, silicon and nitrides element (named as SiBCN) was synthesized from liquid ceramic precursor method. Besides, the pyrolysis, microstructure, and chemical composition were studied at 1600 ℃. The results showed that the samples with different boron content have similar structural composition and both of the two precursors have stable SiBN amorphous structures at 1400 ℃,which are mainly composed of B-N and Si-N and endows them with excellent thermal-oxidation stability. With the progress of the heating process, the more boron content and the more amorphous structures, which significantly improves the thermal stability of the samples in extreme high environment. However, during the moisture treatment, introduction of more boron leads to worse moisture stability.   “

Q4: The introduction is very broad and weak and requires additional information, in particular previous studies.

Responds:

Thank you for your reminder, we have carefully revised the manuscript and added citations and additional information.

Round 2

Reviewer 2 Report

The current revised version of the paper has been significantly improved. Several parts in the manuscript have been re-written and the English has been improved, which adds to the readability of it. There are some details remaining, for example the different shape of the 900oC curve in figure 15 compared to figures 7 & 11, which I do feel that affects strongly the understanding of the paper.

Comparing Figures 7, 11 & 15, it is observed a totally different shape for the 900oC curve

Author Response

Dear Editors:

Enclosed please find the manuscript “Experimental investigations into the pyrolysis mechanism and composition of ceramics precursor containing boron and nitrides with different boron content” by YiQiang Hong, et al.. Some revisions and explanations have been made in accordance with reviewers’ suggestions for the manuscript.

The corresponding revisions were made in sequence as shown below.

Reviewers ' comments and the responds of authors:

Reviewer: 2

Q1:The current revised version of the paper has been significantly improved. Several parts in the manuscript have been re-written and the English has been improved, which adds to the readability of it. There are some details remaining, for example the different shape of the 900oC curve in figure 15 compared to figures 7 & 11, which I do feel that affects strongly the understanding of the paper.

Comparing Figures 7, 11 & 15, it is observed a totally different shape for the 900oC curve

Responds:

       Thank you for your patient reply. My discussion of the experiment may have caused your misunderstanding. Next, I will explain the differences in the data to you in detail of Fig15-900℃.

The initial products derived from two of the liquid precursor pyrolysis with the heat treatment at 900 ℃ in NH3. As described in the paper The products were named as samples-900N. based on these samples-900N, we carried out further experiments in turn.

As show in the Figure 7, samples-900N were heated at 900℃, 1200℃, 1400℃, 1600℃ again, but in this heat treatment, the environment is N2. Figure 11 provides the experimental data of the samples with heat treatment at 900℃, 1200℃, 1400℃ in air, at the same time, the results of the correlational analysis about the samples with heat treatment at 900℃, 1200℃, 1400℃ in vacuum are presented in Figure 15. Hence, it can be clearly seen form the above discussion that the differences of the 900℃curves derived from different heat treatment methods, that is, the 900℃curve presented in Figure 7, Figure 11 and Figure 15 came from three different samples.